# Current State of Liver-Directed Therapies and Combinatory Approaches with Systemic Therapy in Hepatocellular Carcinoma (HCC)

**DOI:** 10.3390/cancers11081085

**Published:** 2019-07-31

**Authors:** Pedro Viveiros, Ahsun Riaz, Robert J. Lewandowski, Devalingam Mahalingam

**Affiliations:** 1Developmental Therapeutics, Robert H. Lurie Comprehensive Cancer Center, Northwestern University Feinberg School of Medicine, Chicago, IL 60611, USA; 2Department of Radiology, Northwestern University Feinberg School of Medicine, Chicago, IL 60611, USA

**Keywords:** HCC, hepatocellular carcinoma, liver-directed therapy, TACE, radiation, RFA, Yttrium-90, combination, VEGF, immunotherapy

## Abstract

The increasing set of liver-directed therapies (LDT) have become an integral part of hepatocellular carcinoma (HCC) treatment. These range from percutaneous ablative techniques to arterial embolization, and varied radiotherapy strategies. They are now used for local disease control, symptom palliation, and bold curative strategies. The big challenge in the face of these innovative and sometimes overlapping technologies is to identify the best opportunity of use. In real practice, many patients may take benefit from LDT used as a bridge to curative treatment such as resection and liver transplantation. Varying trans-arterial embolization strategies are used, and comparison between established and developing technologies is scarce. Also, radioembolization utilizing yttrium-90 (Y-90) for locally advanced or intermediate-stage HCC needs further evidence of clinical efficacy. There is increasing interest on LDT-led changes in tumor biology that could have implications in systemic therapy efficacy. Foremost, additional to its apoptotic and necrotic properties, LDT could warrant changes in vascular endothelial growth factor (VEGF) expression and release. However, trans-arterial chemoembolization (TACE) used alongside tyrosine-kinase inhibitor (TKI) sorafenib has had its efficacy contested. Most recently, interest in associating Y-90 and TKI has emerged. Furthermore, LDT-led differences in tumor immune microenvironment and immune cell infiltration could be an opportunity to enhance immunotherapy efficacy for HCC patients. Early attempts to coordinate LDT and immunotherapy are being made. We here review LDT techniques exposing current evidence to understand its extant reach and future applications alongside systemic therapy development for HCC.

## 1. Introduction

Hepatocellular carcinoma (HCC) is the fourth leading cause of worldwide cancer death [1]. An important part of the prognosis of HCC lies on the underlying chronic liver diseases that are also risk-factors for its occurrence. A Child–Pugh score for cirrhosis mortality is essential in the stratification of patients. In addition to the number and size of tumor and patient’s performance status it is used to generate the Barcelona Clinic Liver Cancer (BCLC) staging system. BCLC staging system is unique in that it allocates treatments based on the stage. Screening strategies for patients with hepatopathies may have increased the proportion of newly diagnosed patients with localized (BCLC-A) disease [2,3]. Yet, most patients are ineligible to curative surgical treatments due to liver disfunction or disease extension. Approximately 25% of patients present as intermediate stage (BCLC-B), reflecting a bigger size or higher number of lesions [4]. For these, the main treatment modality is liver-directed therapy (LDT) [5]. Further, a subset of patients that present with more advanced BCLC-C stage are also candidates for LDT. 

The varied set of LDTs have become an integral part of HCC treatment. They are currently used for a wide range of purposes, from local control and symptoms palliation to bold curative strategies. Local ablative strategies are generally used to treat small tumors (up to 3 cm) in favorable locations. Tumors greater than 3 cm are generally subject to arterial embolization. New external radiotherapy technologies are also being used in both scenarios. The biggest challenge in face of these innovative and sometimes overlapping technologies is to identify the best fit for each case. In medical practice, many HCC patients derive benefit from LDTs used as a bridge to a curative treatment. Radioembolization with yttrium-90 (Y-90) has gained ground over the years, but trans-arterial chemoembolization (TACE) strategies are still the most widely used methods for locally advanced cases. LDTs may also generate opportunity to postpone systemic therapy. Nonetheless, there is expanding interest on LDT-led changes in tumor biology that could impact systemic therapy efficacy. Additional to its apoptotic and necrotic properties, different LDT strategies could influence vascular endothelial growth factor (VEGF) expression and release. However, trans-arterial chemoembolization (TACE) used alongside tyrosine-kinase inhibitor (TKI) sorafenib has had its efficacy contested and, most recently, interest in associating Y-90 and other LDT to TKIs is emerging. 

Furthermore, LDT-led changes in tumor immune microenvironment and immune cell infiltration could be an opportunity to enhance immunotherapy efficacy for HCC patients, which is overall low. Early attempts to coordinate LDT and immunotherapy are being pursued, mostly in patients with advanced HCC. An important question is whether different LDTs have been surveyed for safety and applicability in the pre-transplant scenario. We here review LDT techniques exposing current evidence to understand its extant reach and future applications alongside systemic therapy development for HCC.

## 2. Overview on the Current State of Liver Directed Therapy Strategies

### 2.1. Percutaneous Ablative Approach

Amongst the first-line options for HCC with curative potential there are thermal ablative methods such as radiofrequency ablation (RFA) [6,7,8,9]. RFA has emerged as a viable alternative for patient’s ineligible for liver resection or transplantation (Appendix A). In RFA, an image-guided technique, an electrical current generates a localized rise in temperature that leads to cell death. For very early disease. A comprehensive meta-analysis compared the different types of ablation usually employed in early-stage HCC with RFA [6]. No significant complete tumor ablation, local tumor recurrence or overall survival (OS) rate differences were encountered between patients who were submitted to RFA and microwave ablation (MWA). Similar complication rates were also described [6] MWA relies on the delivery of high-frequency microwave to the tumor bed. It has the advantage of taking less time to perform as well as the possibility of multiple ablations at the same time [7]. 

When comparing RFA and percutaneous ethanol injection (PEI; a percutaneous chemical ablative method), despite the significantly higher overall response (OR) for complete tumor ablation when PEI was used, one and three-year local tumor recurrence rates were higher in this group (odds-ratio 2.25, 95% CI 1.15–4.83 and 2.44 95% CI 1.10–5.41, respectively), maybe reflecting the inability to ablate margins with this technique. The complication rates were equivalent [6]. Although inferior, PEI may still be an alternative in small tumors for which RFA is not an option [6,10,11,12,13]. Examples are tumors in the proximity of gallbladder or major vessels [14]. 

Cryoablation is dependent on a cryoprobe that makes it possible to freeze the tumor and cause irreversible damage [15]. In the same metanalysis, RFA was compared with cryoablation [6]. There was a tendency for the latter to be associated with higher local tumor recurrence in the retrospective analyses (odds-ratio 2.10, 95% CI 0.65–6.78) but not in the randomized trials (OR 0.53, 95% CI 0.24–1.18). In the subgroup of tumors > 2 cm, cryoablation presented lower recurrence rates. Reported complication did not differ significantly between the two methods [6], but case reports of liver fracture have limited its widespread application. 

There is still not enough data on high intensity focused ultrasound (HIFU) for HCC treatment, for lack of comparison with established ablation methods [6]. Irreversible electroporation was not addressed in the metanalysis. It causes irreversible breaks to the tumor-cell membranes by high-intensity electrical pulses. It is offered as an option for tumors in the proximity of blood vessels and bile ducts [16]. Another ablation method, laser thermal ablation was found to have similar outcomes with a tendency towards lower complication rates when compared with RFA [6,17]. To the best of our knowledge, randomized trials to support these methods are currently lacking.

RFA has been used in HCC of up to 5 cm sized tumors. An ample retrospective study (*n* = 1894) utilizing the SEER database was planned to evaluate differences in outcome after RFA, liver resection and transplant strategies. In the group of patients with tumor size ≤ 2 cm, and in the range of 2–3 cm, no OS difference was identified when comparing RFA and resection (median OS 70 vs 60 months *p* = 0.10; OS 70 vs 61 months, *p* = 0.70). In tumors > 3 cm, RFA was inferior to resection in OS (median OS 65 vs 49 months *p* < 0.01) and disease specific survival [18]. It is important to note that patients who underwent resection likely had improved liver function when compared to the patients who underwent percutaneous ablation. Transplant strategies were consistently superior in all cohorts. From a retrospective standpoint, RFA seems to be equivalent to surgical resection strategies specifically in smaller tumors, being a valid alternative for selected cases [18]. For tumor bigger than 20–30 mm, there is a tendency to try associating ablation with some type of arterial embolization, most commonly TACE, although there is a general lack of randomized clinical trials to support that [19,20].

### 2.2. Stereotactic Body Radiation Therapy

SBRT has been attempted in patients with small, inoperable tumors ineligible or refractory to standard ablation therapies. A cohort (*n* = 34) of HCC patients, 84% refractory to previous techniques, underwent SBRT. The majority received 45 Gy in three or five fractions. One-year local control was 94%. One-year OS was 84%. SBRT provided local control with a manageable side effect profile. Altogether, 19% of the patients experienced worsening of Child–Pugh score by 2 or more points, usually temporary, and 34% presented grade 3 or greater toxicities [21]. A second cohort of 37 HCC patients with 43 lesions and a median lesion diameter of 2.7 cm underwent SBRT, median dose of 50 Gy, with only one local failure after a 14 months median follow-up. All Child–Pugh B patients underwent a one-month break after 3 fractions to assess hepatic toxicity, with only 1 out of 9 of these patients presenting with grade 3 or greater liver toxicity, delineating a potential toxicity-sparing strategy for these patients [22]. A randomized phase III trial (NCT03898921) of RFA vs SBRT is currently recruiting small (<5 cm) HCC patients. 

Moreover, proton radiation therapy, a novel external radiation strategy, has gained ground in the attempt to curb hepatic toxicity and liver failure following treatment. It was correlated with a decline in the risk of non-classic radiation-induced liver disease (OR 0.26, *p* = 0.03), possibly associated with a significantly higher OS (two-year survival rate of 59.1 vs 28.6%, HR 0.47, *p* < 0.01), when compared with the better-established photon ablative radiation therapy with no difference in regional recurrence [23]. The development of new external radiation techniques with lower rates of serious adverse events has broadened its use from the original palliative intention to definitive/curative and “transplant bridging”[24]. 

For intermediate stage (BCLC-B) disease with larger HCC lesions post-TACE viable disease is not uncommon. External radiation is emerging as an option. A retrospective analysis of 209 patients receiving TACE or SBRT for 1 or 2 nodules, found no difference in OS and favorable local control rates for SBRT [25]. Despite the increase in interest and utilization of SBRT in HCC, there is general a lack of randomized trials. A phase III trial (planned n = 120) is investigating the role of SBRT against repeated TACE in HCC patients with incomplete response to previous TACE sessions (NCT02921139) [26].

### 2.3. Trans-Arterial Treatment Techniques

In intermediate stage (BCLC-B) patients (where curative strategies are not possible) trans-arterial techniques using different embolization strategies are usually the treatment of choice [27]. They take advantage of the special blood supply of the liver. HCC is mainly supplied by the hepatic artery while the surrounding tissues can be sustained by the portal venous system alone. The development of strategies to embolize the tumor and instill chemotherapy or even radioactive particles in the artery system that support the tumor has created treatment options for those that are to large or numerous to be treated with ablation. Usually, availability and experience in different institutions/health systems and patient-specific technicalities are important factors when selecting the modality. Defining the patient who benefits from a specific embolization technique is a complex matter.

Different macro-embolic therapies were designed to block tumor blood supply (Appendix A). Bland particle embolization (TAE) involves delivering small particles to the tumor to induce ischemia/hypoxia [28,29]. cTACE (conventional TACE) is the emulsion of aqueous chemotherapy in lipiodol, typically followed by bland embolization to prevent washout of drug [30]. Chemotherapy with doxorubicin or cisplatin is commonly used. It is common for these patients to experience self-limited toxicity, with the peak of occurrence happening within several days. Full-recovery is expected in one or two weeks. In the postembolization syndrome, fever, local pain, and nausea are associated with transient elevation of bilirubin, aspartate aminotransferase, and alanine aminotransferase levels. Rarer but important local side-effects are liver rupture (extremely rare), formation of liver abscess and bilomas (rare complication which happens in patients who have had prior ampulla violating procedures such as biliary stents). Femoral artery pseudoaneurysm and pulmonary embolism are also reported [31].

TACE is the sole LDT method with proven survival edge over best supportive care for advanced and unresectable HCC patients who cannot be treated with ablation or transplant [32,33]. TACE is an adequate strategy for patients with tumors without branch portal vein thrombosis. 

TACE has seldom been compared with other embolization procedures. A metanalysis on 5 retrospective studies that compared TACE to bland particle embolization (TAE) in HCC patients found a non-significant 2-year survival rate favoring TACE (RR 0.88, *p* = 0.38). The common post arterial embolization symptoms were similar between groups. Nausea, vomiting, and transient hematological toxicity were more common for TACE patients [34] reflecting the systemic effects of chemotherapy agents employed. TAE is also a less costly strategy.

A special form of TACE, drug-eluting bead TACE uses local drug-eluting beads have been applied aiming to enhance the local delivery of chemotherapeutic agents to the tumor without increasing toxicity [35,36,37]. Drug-eluting bead TACE was compared with conventional TACE in a retrospective analysis (*n* = 250). The OS was comparable between groups (12.3 vs 13.6, respectively, *p* > 0.05) despite the lower average number of sessions for drug-eluting bead TACE (median 2.9 vs 4.0 months, *p* = 0.01) [38].

### 2.4. Radioembolization

In trans-arterial radioembolization microspheres are impregnated with yttrium-90 (Y-90) and are directed through the tumor blood supply [39]. The arterial hypervascularity of HCC allows for preferential tumoral uptake of these particles leading to high doses of radiation without causing ischemia [40]. Patients who are found at risk of lung shunting leading to ≥30 Gy to the lung per treatment session and ≥50 Gy cumulatively (multiple treatment session) should not receive Y-90. This lung shunting is generally determined by a pre-treatment technetium-99m macroaggregated albumin scan. General specification on liver function are analogous to the parameters used before TACE. The advantages over TACE include relatively low-toxicity and possibility of utilization in patients with portal vein thrombus (PVT) as this is not a macroembolic technique and does not lead to arterial hypoperfusion [41,42]. Also, Y-90 is usually done in fewer sessions when compared to TACE [43]. 

The indications of Y-90 have expanded beyond BCLC B patients. Patients who are within the Milan criteria are bridged to liver transplant given long wait times for liver transplants [44]. Patients who are beyond the Milan criteria may be down-staged to within transplant criteria and become eligible for liver transplants. Radiation segmentectomy where high doses can be administered to less than or equal to 2 hepatic segments can prove curative and may be considered ablative. Radiation lobectomy is a technique that can be performed in HCC to treat the tumor and cause radiation fibrosis in the lobe with the tumor to induce contralateral lobe hypertrophy so patients can become candidates for surgical resection [45]. Patients with PVT (BCLC C) have been successfully and safely treated with Y-90 [46]. Y-90 has also been used as a palliative technique in patients with large tumors causing pain.

For Y-90, there was evidence that time-to-progression is at least equivalent to TACE, with improved toxicity rates with Y-90 [47,48,49]. There is currently no prospective study supporting its effects on survival. A meta-analysis including nine observational studies and 2 randomized trials (*n* = 1652) found an association of Y-90 with higher OS, response rates and lower adverse event rates when compared to conventional TACE. Symptoms of post-embolization syndrome are less expected [50]. The first randomized phase II, with (*n* = 28) Child–Pugh A patients identified similar disease control rates between the treatment strategies [51]. In a second trial (*n* = 45), longer median time to progression with Y90 (>26 vs 6.8 months, *p* < 0.01) when compared with the group receiving conventional TACE, with similar response rates [52]. A second meta-analysis compared randomized trials for different treatment strategies for unresectable HCC, including different trans-arterial embolization and local ablation techniques. Fifty-five randomized clinical trials (*n* = 5763) were analyzed, including 12 direct comparisons. With a moderate quality-of-evidence, TACE and DEB-TACE were not associated with better overall survival (OS) rates when compared with bland particle embolization (Median OS 18.1, 20.8, and 20.8 respectively). Trans-arterial radioembolization with Y-90 did not confer a survival advantage over TACE strategies, but with a low quality-of-evidence. However, it was identified as the safest embolization strategy [53].

### 2.5. Sequencing

When considering TACE as a treatment option, scoring systems could be of help in the decision-making process [54,55,56]. Although these scoring systems may not be validated for similar treatment strategies, the best fit patients for trans-arterial radio- or chemoembolization are the ones with preserved liver function as well [47]. There is concern over the modernization of BCLC-B and C classification to differentiate for different LDT indications, contraindication and outcomes and best define the best fit strategy for each patient [57].

## 3. Liver Directed Therapies Role Pre-Operatively

Trans-arterial embolization techniques may be used in pre-operative scenarios, with most reported experience using TACE to downstage or downsize tumor prior to planned surgical resection [58,59]. In one single center experience (*n* = 67) patients who underwent bland trans-arterial embolization or TACE, 87% had a pathological response. Twelve percent of the patients had a downstaging to fit Milan criteria [58,59].

For Y-90, there is no prospective trial comparing it with TACE in this specific scenario. In a retrospective study involving nine institutions (*n* = 47), the most common histology being HCC, it was demonstrated that hepatectomies are usually safe following Y-90. The complication rates were similar to those of other hepatectomy cohorts. The 90-day complication rate was 43% and mortality rate was 2% [60]. 

Hepatectomy following Y-90 seems to be a safe strategy in select populations. Y-90 has been used for tumor control and contralateral lobe hypertrophy prior to surgical resection in HCC patients [61]. TACE may also be a utilized prior to PVE if a major liver resection is expected; particularly before right side resections [62]. Higher complication rates were associated with previous bilobar Y-90 treatment, OR 4.5, revealing a potential limitation to its application before hepatic resections [60].

## 4. Liver Directed Therapies Role in Bridging Orthotopic Liver Transplant

In early-stage HCC, usually superimposed on chronic liver disease, transplant addresses not only the tumor but also the underlying condition (2). For patients who meet Milan criteria the time to liver transplant is challenged as a prognostic factor [63]. However, despite the points system used to select the patients in the United States, the waitlist is usually high enough to consider a bridging therapy. The concept behind bridging therapy is to prevent patients from dropping off the transplant list; locoregional therapies are employed to maintain tumor number/size within accepted Milan criteria while waiting for liver transplant. Additionally, mandatory waiting time is now 6 months and the use of bridging therapies is widespread. The main objective in this scenario becomes generating a window of disease control without prohibitory toxicity. The plurality of LDTs have been employed in this scenario. Choosing the better applicable LDT for each patient is very important [18].

In a retrospective study (*n* = 3601) liver transplant recipients who did not receive LDT had similar recurrence free survival rates to the ones who underwent any previous bridging therapy (1, 3, 5-year rates of 89%, 77%, and 68% vs 85%, 75%, and 68%; *p* = 0.49). There was no difference in 5-year post liver transplant recurrence (11.2% vs 10.1%, *p* = 0.47). Increased LDT numbers and unfavorable waitlist alpha-fetoprotein trend significantly predict recurrence. LDT modality did not seem to influence recurrence [64]. Patients with complete pathological response in the explant liver had a superior 5-year recurrence free survival and lower recurrence rates when compared with patients who either did not receive LDT before transplant and those who did not achieve complete pathological response (all *p* < 0.05) in multivariate analyses controlling for pre-transplant variants [64]. 

Many treatment strategies here previously discussed have been successfully attempted before liver transplantation in order to control tumor growth and even to venture tumor downstaging [65]. TACE is perhaps the most studied form of liver-directed therapy used as a bridge to liver transplant and is ubiquitous in the guidelines. Newer and less studied then conventional TACE, drug-eluting beads TACE employs an innovative technology combining arterial embolization and locally eluting chemotherapy. The complete pathologic response rate from explanted liver (*n* = 111) was higher than 50% for both conventional TACE and drug-eluting beads TACE (50.7% and 57.1%, respectively) [65]. In a retrospective study of 34 patients that underwent doxorubicin drug-eluting bead TACE and orthotopic transplant when compared with 60 patients who did not underwent TACE before transplant, there was no statistically significant difference in clinical outcomes [66]. RFA is also widely used, but usually limited to small tumors. 

As previously mentioned, trans-arterial embolization techniques usually require an adequate liver function. Although a preserved liver function is also required for SBRT it presents itself as a viable third option. There is limited information on its efficacy in this pre-transplant setting. An intention-to-treat analysis was executed in a cohort of 379 patients treated with one of these LDT. SBRT (*n* = 36), TACE (*n* = 99) or RFA (*n* = 244). The drop-out rate was 16.7%, 20.2%, and 16.8%, respectively (*p* = 0.4). Postoperative complication rates were similar between groups. RFA treated patients presented higher rates of tumor necrosis. The 5-year actuarial survival rates were 61%, 56%, 61% (*p* = 0.4). Five-year survival rates from the time of the transplant were 75%, 69%, and 73% (*p* = 0.7) [67].

For Y-90, one retrospective analysis (*n* = 83) compared radioembolization with Y90 and TACE in the scenario of bridging/downstaging to liver transplantation. Partial response was 61% with Y-90 and 37% with TACE. Downstaging was 58% with Y-90 and 31% with TACE. Event free survival favored Y-90 (17.7 vs 7.1 months, *p* < 0.01) [68]. Another retrospective study with (*n* = 172) HCC patients, 93 and 79 were subject to bridging Y90 and TACE before transplant. With a median post-transplant follow-up of 26 months, 9% of the Y-90 and 8% of the TACE patients had recurrent disease. Recurrence-free survival was 79 vs 77 months, respectively *p* = 0.84. Revealing similar post-transplant outcomes. However, Y-90 patients endured significant longer time to transplant 6.5 vs. 4.8 months (*p* = 0.02) when compared with the patients who underwent TACE [69]. Y-90 has been established as a useful bridging modality by the PREMIERE trial, due to its longer median time to progression (>26 months vs 6.8 months, *p* < 0.01) when compared with cTACE [52].

## 5. Role of Liver-Directed Therapies in BCLC-C Patients

Although BCLC-C patients have been included in Y-90 trials, the diversity of patients in this classification needs to be scrutinized [52,70]. The role of Y-90 in BCLC-C patients was investigated from a single-center experience (*n* = 541 BCLC-C patients), Child–Pugh A patients were found to have a median OS of 15 months and 8 months in Child–Pugh B [44]. In another single-center experience (*n* = 69), median OS was 6.1 months. In a multivariate analysis disease burden, ascites and Child–Pugh were prognostic in these patients. In a prospectively evaluated BCLC-C cohort (*n* = 547) undergoing Y-90 radioembolization the median OS was 10.7 months. When accounting for the patients classified as BCLC-C by their ECOG performance status only, the median OS was 19.4 months. Patients with PVT/metastases had a median OS of 7.7 months. On a multivariate analysis, portal vein thrombosis was a significant prognostic factor within BCLC-C classification (*p* < 0.01) [71]. 

Y-90 has been historically performed in disease restricted to the liver with portal vein invasion/thrombosis, occupying the gap created by TACE contraindication in this scenario. In a retrospective analysis (*n* = 29) including BCLC-B (*n* = 18) and BCLC-C (*n* = 11) patients after radioembolization with Y-90, the median OS was 17 months. The majority of patients were Child–Pugh A and 12 of the patients had a diagnosis of portal vein thrombosis. In this small group of patients, portal vein thrombosis was not a significant prognostic factor [41]. In a series of cases, 4 patients had a complete regression of the vascular invasion after receiving Y-90, had a downstaging and were put back to the transplant waitlist [72]. In BCLC-C patients, LDT have not been shown to be superior to standard-of-care systemic therapy [73,74]. For a select group of patients with liver dominant disease it could be an option. 

## 6. Molecular Changes Associated with Liver Directed Therapies.

The vascular endothelial growth factor (VEGF) is a well-known biomarker of tumor angiogenesis and it is directly involved in tumor growth and metastatic spread in HCC [75]. For the last decade VEGF has been explored as a new molecular tool to better predict HCC prognosis, vascular invasion, recurrence free survival post liver transplant and perhaps guide liver-directed therapies [76,77,78]. In a previous meta-analysis with 782 HCC patients, higher serum VEGF levels were associated with poor overall and disease-free survival [79]. It has been postulated that stimulation of angiogenic factors by LDTs could lead to tumor growth. Recently, a biopsy-matched study showed increased expression of hypoxia markers such as VEGF in the subgroup of patients treated with TACE when compared to controls (*p* = 0.046), suggesting how the hypoxic environment induced by TACE could stimulate tumor growth [80]. Moreover, in-vitro HCC models that underwent hypoxic condition mimicking RFA has demonstrated that lower exposure to the hypoxia inducible factor (HIF)-1α signaling significantly reduced invasive and chemo resistant potential of these cells [81]. This evidence supports addressing the benefit of combined anti-angiogenic systemic therapies such as sorafenib with TACE or RFA to minimize disease progression among partial responders with LDT and systemic therapy combinations [82,83].

Conversely, LDT cause tumor disruptions that could liberate tumor antigens and induce cytokine immune response. There is mounting evidence higher infiltration of cytotoxic T-cells and expression of PD-L1 in tumor cells may result in better prognosis in HCC. Higher perivascular tumor infiltrating lymphocytes in HCC correlated with better disease-free survival [84]. The expression of LAG-3 and PD-L1 that promote CD8 T-cell tolerance, are increased in most HCC tumors [84]. An experiment with (*n* = 90) resected specimens of HCC were analyzed by flow cytometry. HCC presenting PD1-high CD8+ T-cell lymphocytes expressing TIM3 and LAG-3 and produce low levels of TNF and interferon-gamma. These cells were incubated and targeted by antibodies acting against PD-1, TIM3, and LAG3. This resulted in higher production of TNF and cell proliferation [85]. Patients undergoing radiofrequency assisted resection had a significant decrease in circulating inhibitory Treg cells (*p* < 0.01) and a significant increase in CD8⁺ T lymphocytes (*p* = 0.05) and CD4⁺ memory T cells (*p* < 0.01), which are important in the immune response exacerbated by utilizing anti-PD-L1 immune checkpoint inhibitors [84]. The immune profile of surgically resected HCC downstaged by Y-90 exhibited higher infiltration of CD8^+^ T cells as well as higher expression of two immune markers, granzyme B and TIM-3 when compared with control [86]. These findings create a leeway for association of LDT and immune-checkpoint inhibitors (Figure 1).

## 7. Current State of Liver Directed Therapies Associated with TKIs

In the incurable scenario, liver-directed therapies have been widely used, sometimes successfully postponing the initial use of systemic therapy [87]. Sorafenib is a tyrosine kinase inhibitor (TKI) whose anti-VEGFR properties were proven active against HCC [88]. It is currently the most widely-used and studied systemic therapy. However, its efficacy is at present suboptimal and its toxicity, especially in patients with liver dysfunction [89], offers some additional reason for the use of LDTs to be stretched. 

Recently, another TKI lenvatinib [90] was approved for unresectable HCC in treatment naïve patients. In the scenario of systemic therapy upon failure of sorafenib, regorafenib which is structurally similar to sorafenib was approved [91] followed by cabozantinib [92] which also target MET and AKL pathways alongside VEGF inhibition. Most recently, ramucirumab, an anti-body targeting type-2 VEGFR was approved in sorafenib-refractory patients with high alpha-fetoprotein [93]. 

Research is currently aimed at identifying unique pathologic and molecular targets in HCC. Different signaling cascade molecules including Wnt, MET, FGFR, VEGFA, MCL1, the EGFR/Ras/MAPK, as well as immune checkpoint proteins CTLA-4, PD-1, and PD-L1 have been identified as therapeutic targets [94]. Alterations in Wnt/ß-catenin signaling pathway seem to play a relevant role in initiation and progression of HCC. Activating mutations in the *CTNNB1* that encodes ß-catenin are present in 15–25% of HCC. Inactivating mutations in *AXIN1*/*AXIN2* genes, encoding an important component of the ß-catenin destruction complex are found in a similar proportion of patients [95]. *CTNNB1* mutant HCC could be targeted by mTOR inhibition [96]. Novel Wnt/ß-catenin inhibitors are undergoing preclinical and clinical evaluation solid tumors including HCC [95].

Despite high response rates after LDTs, some patients experience short-lived disease control. In the period post-embolization, hypoxia is a driver of VEGF expression. Neoangiogenesis may then create the perfect environment for disease to progress after TACE (Figure 1). In the SHARP study the subgroup without extrahepatic disease benefited the most from sorafenib. It was then hypothesized that blocking the angiogenesis pathway after TACE would suppress tumor recurrence. Post-TACE, a phase III study (*n* = 458) comparing sorafenib and placebo after TACE was conducted in patients who responded to TACE. The primary endpoint time to progression (TTP) was not met. The median treatment duration in the experimental group was 17 weeks. There was criticism over the median time to start on sorafenib and placebo (9 weeks) [97]. 

The SPACE trial (*n* = 307), a randomized phase II, compared the combination of doxorubicin-eluting beads TACE with sorafenib with the DEB-TACE alone in Child–Pugh A unresectable and multinodular HCC. Primary endpoint of TTP was similar between groups (169 vs 166 days, HR 0.797, *p* > 0.05), although the time to untreatable progression differed in favor of the sorafenib combination. Secondary endpoint (overall survival) was not reached. In this study, scheduled TACE was performed at fixed intervals, generating the hypothesis that it might have not always been needed. Its excess toxicity might have increased the side effect of sorafenib, what could have influenced the median sorafenib use duration in the experimental group of 21 months [98]. In the phase III trial TACE 2 (*n* = 313), patients were randomized to receive or sorafenib plus TACE or TACE alone. Sorafenib would be withheld at the sign of new intrahepatic lesions. There was no evidence of difference in primary-endpoint PFS between groups (238 vs 235 days, HR 0.99, *p* > 0.05) [99].

These studies have in common a short sorafenib treatment duration in the experimental group (Table 1). Additionally, by using RECIST 1.1 to define progression, many patients with a single new liver lesion were considered to have progression. This new lesion could most of the times be addressed by TACE. Thus, some would argue it was not a good definition of disease progression when using TACE in combination. The TACTICS trial (*n* = 156) was a randomized trial comparing TACE plus sorafenib with TACE alone in unresectable patients without distant metastasis. All patients were Child–Pugh ≤7, had a maximum of two previous TACE sections, and presented with up to 10 nodules not exceeding 10 cm. This study utilized a new un-TACE-able progression-free survival (PFS) that considered a ≥25% increase in viable lesions, decline of liver reserve to Child–Pugh C, development of extra-hepatic metastasis or vascular invasion or becoming TACE-refractory. The primary endpoint of PFS was reached (25.2 months vs 13.5 months, *p* < 0.01) favoring the combined approach arm. The initial response rates did not differ between arms. The interval between TACE sessions was significantly longer in the combination arm. The median duration of sorafenib in the combination arm was 38.7 months [100]. The efficacy of associating of TKIs and LDTs [101] has yet to be proven. Although, there is another ongoing phase III clinical study (NCT03905967) to evaluate if adding TACE with lenvatinib Is better than lenvatinib alone as first-line therapy in advanced HCC, the enthusiasm of the combination strategy is waning in particular amongst patients with advanced BCLC-C HCC where growing number of systemic therapy options are emerging.

## 8. Yttrium-90 vs. Sorafenib

Two prospective randomized trials comparing radioembolization and Sorafenib, the SARAH (Sorafenib versus radio-embolization in advanced hepatocellular carcinoma) and SIRveNIB (study to compare selective internal radiation therapy versus Sorafenib in locally advanced hepatocellular carcinoma) trials were negative [73]. The SARAH trial, open for BCLC-C patients, aimed to compare the efficacy of Y90 with the standard-of-care sorafenib. The 467 patients were randomized in two therapy groups with an approximate median follow up of 28 months. OS was 8.0 months with Y-90 and 9.9 months in the sorafenib group (*p* = 0.18) [74]. In both trials, although the ORR was significantly better in the Y-90 groups and Y-90 was better tolerated than Sorafenib; there was no statistically significant difference in the primary endpoint of OS. addition, there was no significant difference in survival in the subgroup of patients presenting with vascular invasion (Table 2). These trials present some limitations. In these trials, many patients didn’t actually receive radioembolization and there was a longer time to intervention in the radioembolization arm. Most study sites had limited experience with radioembolization, potentially limiting safety/therapeutic outcomes. Also, patient selection was problematic, as patients who failed TACE, potentially limiting arterial supply to tumors, and those who had main PVT (a relative contra-indication to Y-90) were included. Further, with current evidence, comparing systemic therapy to LDT is suboptimal as there is accumulated evidence of benefit of systemic therapy over best supportive care [88,90]. Combining systemic and loco-regional therapies is more of interest. 

When the combination of Y-90 with sorafenib was evaluated retrospectively from a single-center experience the progression-free survival was greater than what historically described with sorafenib [102]. Similar to studies with TACE and systemic therapy combination, sequential LDT followed by systemic agents are still practiced widely, its true benefit remains unclear. Careful patient selection for these strategies are advised. 

## 9. Role Liver Directed Therapies Associated with Immunotherapy

Anti-PD1 antibody nivolumab and pembrolizumab, are immunotherapeutic agents approved for HCC, following failure or patient unfitness to receive TKIs. However, it is still only efficacious to an undefined subgroup of patients [103,104]. Different immunotherapeutic agents are currently under investigation as anti-HCC agents. Finding co-stimulatory targets to increase immune response has been a strategy to mitigate the low efficacy of PD-1 inhibition for the majority of HCC patients. Glucocorticoid induced TNFR-related protein (GITR) is under clinical investigation for a variety of solid tumors. Its expression is highest in in activated Treg lymphocytes but is still found in effector T cells. For HCC, a proof-of-concept pre-clinical experiment demonstrated that agonistic targeting of GITR with a GITR ligand or activating GITR antibody amplified the ex-vivo activity of tumor-infiltrating T-cells in HCC. This was illustrated by CD8+ T cell proliferation and granzyme B expression [105]. Targeting GITR and PD1 may improve anti-cancer T-cell in some patients and GITR is a promising immunotherapy target in HCC.

In theory, LDTs could lead to higher neoantigen presentation and be a stimulus to immune response. Clinical success stories when adding anti-PD-1 to LDT like Y90 [106,107] and TACE have only amounted to case reports to the best of our knowledge. Clinical trials are warranted to explore how LDT may influence the response to immunotherapeutic agents. 

Our institution is currently conducting a phase I trial (NCT02837029) in advanced HCC patients to receive Y-90 and nivolumab, in patients that are either treatment naïve or has had systemic therapy but deemed candidate for LDT. Preliminary data on safety is encouraging and efficacy data is pending. Similarly, we are evaluating this combination (NCT03812562) of radioembolization with Y-90 and nivolumab in the pre-surgical scenario. A single-arm phase II trial (NCT03572582) is currently recruiting multinodular intermediate stage HCC patients to test the combination of TACE and nivolumab. A similar phase 2 study (NCT03143270) is recruiting advanced HCC patients to receive drug-eluting TACE alongside nivolumab. Finally, a phase II (NCT03099564) is open for advanced HCC patients to undergo pembrolizumab 200 mg IV every 3 weeks in conjunction with Y-90 radioembolization performed one week after the first dose of pembrolizumab.

Enthusiasm surrounding immunotherapy in HCC, has led to multiple studies utilizing checkpoint inhibitor combinations with TKIs or LDT in various stages of HCC. Immunotherapy is currently under conditional approval based on phase II studies. It remains to be seen if this strategy works across HCC patients or if there will be selective patient enrollment guided by biomarkers in the future. 

## 10. Conclusions

LDTs are widely used for HCC. In early stage, a diverse range of ablation techniques are available. RFA is the dominant strategy due to its more established use. Other strategies are still valuable for technically challenging cases, such as tumors in close contact with vessels, or as an alternative to RFA when it is not available. For intermediate stage, TACE is the most widely recommended method of embolization but generalization may prevent the selection of the best strategy for each case. There is compelling evidence that most embolization strategies are at minimum similarly efficacious. Also, trans-arterial radioembolization with Y-90 could be more tolerable. For patients with a lobar branch involvement of portal vein thrombosis Y-90 seems to be the way to go. Pre-operative use of the different embolization strategies seems to be safe and aiming to bridge patients within Milan criteria or downstage patients into Milan Criteria for orthotopic liver transplantation may be successful for a select group of patients. Although there is no evidence of superiority in postponing systemic therapy with TACE and Y-90 in BCLC-C cases with liver-limited disease from randomized studies, it is a valid strategy for a select group of patients. Newer external radiation technologies are being brought to the front row of options for early and intermediate stages. The combination of TACE or other LDTs with systemic therapy is still under scrutiny but there is new light after the untraceable study. When trying to make immunotherapy work better in HCC there is willingness over adding LDTs and there is a biological rationale. The evidence of clinical benefit is still lacking. In intermediate stage HCC, the heterogeneity of current staging systems does not differentiate adequately to guide for the best strategy, leaving a lot of nuances on the hands of HCC-treating specialists. For many of the questions unresolved, new light is expected from ongoing randomized clinical trials.

## Figures and Tables

**Figure 1 cancers-11-01085-f001:**
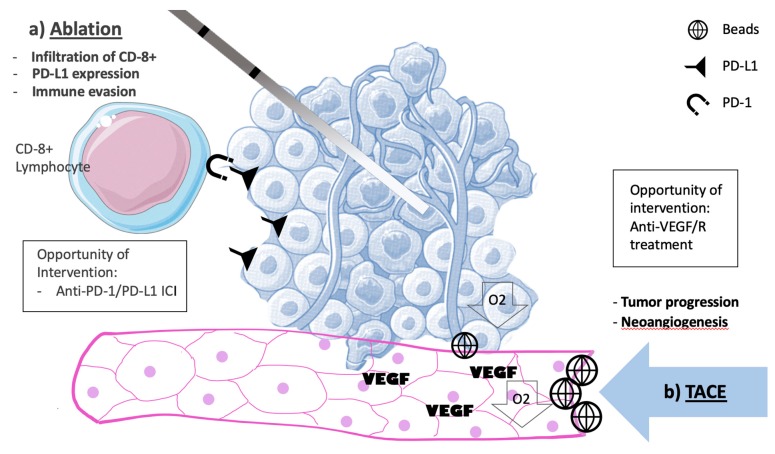
Liver directed therapies and changes in the tumor microenvironment. Increased vascular endothelial growth factor (VEGF) and neoangiogenesis with the potential for tumor progression after TACE; Infiltration of CD8+ Lymphocytes with resultant higher PD-L1 expression with the potential for tumor immune-evasion after ablation. RAF, radiofrequency ablation; TACE, transarterial chemoembolization; VEGF, vascular endothelial growth factor; VEGFR: vascular endothelial growth factor receptor.

**Table 1 cancers-11-01085-t001:** Randomized trials evaluating Sorafenib concomitant to TACE. TTP: time to progression; HCC: hepatocellular carcinoma; DEB-TACE: drug-eluting bead TACE; TACE: trans-arterial chemoembolization; RECIST: Response Evaluation Criteria In Solid Tumors; mRECIST: modified RECIST; cTACE: conventional TACE; PFS: progression-free survival; HR: hazard ratio.

Study	Inclusion Criteria	Regimens	Primary Endpoint	Duration of Sorafenib	Reference
SPACE trial (*n* = 307) Phase 2	Unresectable, multinodular HCC	DEB-TACE + sorafenib vs. DEB-TACE	TTP mRECIST 169 vs. 166 days HR 0.80 *p* > 0.05	21 weeks	Lencioni et al. [98]
TACE 2 (*n* = 313) Phase 3	Unresectable	DEB-TACE + sorafenib vs. DEB-TACE	PFS RECIST 1.1 238 vs. 235 days HR 0.99 *p* > 0.05	17 weeks	Meyer et al. [99]
TACTICS (*n* = 156), 2018 Phase3	Unresectable Child-A 10 lesions max 10 cm max	cTACE + sorafenib vs. cTACE	UnTACEable PFS 25.2 vs. 13.5, HR 0.59 *p* < 0.01	39 weeks	Kudo et al. [100]

**Table 2 cancers-11-01085-t002:** Randomized trials comparing transarterial radioablation with systemic therapy. SARAH: Sorafenib versus radio-embolization in advanced hepatocellular carcinoma; SIRveNIB: study to compare selective internal radiation therapy versus Sorafenib in locally advanced hepatocellular carcinoma; BCLC: Barcelona Clinic Liver Cancer; OS: overall survival.

Study	Inclusion Criteria	Regimens	Primary Endpoint	Reference
SARAH (*n* = 467), 2017 Phase3	Unresectable or refractory to TACE or BCLC C	Y-90 vs. sorafenib	OS 8.0 vs. 9.9, *p* < 0.18	Vilgrain et al. [74]
SIRveNIB (*n* = 360) Phase 2	Locally advanced	Y-90 vs. sorafenib	OS 8.8 vs. 10, *p* < 0.36	Chow et al. [73]

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
