# Peer review of "Current State of Liver-Directed Therapies and Combinatory Approaches with Systemic Therapy in Hepatocellular Carcinoma (HCC)"

_cancers, 2019, doi:10.3390/cancers11081085_

Round 1
Reviewer 1 Report
- The manuscript has as it is no connection to mechanistic basis of disease and which may actually link to treatment success. Maybe the authors should make reference to a recent review here, e.g. the recent in Cancers (Wang W, Smits R, Hao H, He C. Wnt/β-Catenin Signaling in Liver Cancers. Cancers (Basel). 2019 Jul 2;11(7). pii: E926).
- The section on immunotherapy lacks discussion on GITR (van Beek AA, Zhou G, Doukas M, Boor PPC, Noordam L, Mancham S, Campos Carrascosa L, van der Heide-Mulder M, Polak WG, Ijzermans JNM, Pan Q, Heirman C, Mahne A, Bucktrout SL, Bruno MJ, Sprengers D, Kwekkeboom J. GITR ligation enhances functionality of tumor-infiltrating T cells in hepatocellular carcinoma. Int J Cancer. 2019 Aug 15;145(4):1111-1124)
- Upon FDA/EMEA registration for hepatocellular carcinoma (HCC), sorafenib received a broader therapeutic indication than the eligibility criteria of the landmark SHARP trial, studies, but it seems that sorafenib usage should be restricted to Child-Pugh A patients ( Labeur TA. Are we SHARP enough? The importance of adequate patient selection in sorafenib treatment for hepatocellular carcinoma.Acta Oncol. 2018 Nov;57(11):1467-1474. doi: 10.1080/0284186X.2018.1479070. Epub 2018 Jun 26.) Maybe it is good to mention this.
Author Response
Thank you for the great suggestions
We have included the role of WNT signaling pathway and also the reference by Wang et al into the text under the section of current state if liver directed therapies associated with TKIs
Similarly the role of GITR to enhance immunotherapy response in HCC and reference article as suggested.
Finally, based on the reviewers suggestion on the use of sorafenib or its limitation especially amongst patients with poor liver function is added to text and reference article the reviewer suggested.
Reviewer 2 Report
I’m very glad to review the paper titled " Current state of liver-directed therapies and combinatory approaches with systemic therapy in hepatocellular carcinoma (HCC).". It was a well-reviewed article. I think this manuscript is suitable for publication in this journal without any revisions.
Author Response
Thank you for the review and your approval of the manuscript as is.